# NeuralMinimizer: A Novel Method for Global Optimization

**Ioannis G. Tsoulos** [1],*, **Alexandros Tzallas** [1], **Evangelos Karvounis** [1] and **Dimitrios Tsalikakis** [2]

1    Department of Informatics and Telecommunications, University of Ioannina, 47100 Arta, Greece
2    Department of Engineering Informatics and Telecommunications, University of Western Macedonia, 50100 Kozani, Greece
*    Correspondence: itsoulos@uoi.gr

**Abstract:** The problem of finding the global minimum of multidimensional functions is often applied to a wide range of problems. An innovative method of finding the global minimum of multidimensional functions is presented here. This method first generates an approximation of the objective function using only a few real samples from it. These samples construct the approach using a machine learning model. Next, the required sampling is performed by the approximation function. Furthermore, the approach is improved on each sample by using found local minima as samples for the training set of the machine learning model. In addition, as a termination criterion, the proposed technique uses a widely used criterion from the relevant literature which in fact evaluates it after each execution of the local minimization. The proposed technique was applied to a number of well-known problems from the relevant literature, and the comparative results with respect to modern global minimization techniques are shown to be extremely promising.

**Keywords:** global optimization; neural networks; stochastic methods





## 1. Introduction

An innovative method for finding the global minimum of multidimensional functions is presented here. The functions considered are continuous and differentiable and defined as $f : S \rightarrow R, S \subset R^n$. The problem of locating the global optimum is usually formulated as:

$$x^* = \arg\min_{x \in S} f(x) \tag{1}$$

with $S$:

$$S = [a_1, b_1] \otimes [a_2, b_2] \otimes \dots [a_n, b_n]$$

A variety of problems in the physical world can be represented as global minimum problems, such as problems from physics [1–3], chemistry [4–6], economics [7,8], medicine [9,10], etc. During the past years, many methods, especially the stochastic one, have been proposed to tackle the problem of Equation (1), such as Controlled Random Search methods [11–13], Simulated Annealing methods [14–16], Differential Evolution methods [17,18], Particle Swarm Optimization (PSO) methods [19–21], Ant Colony Optimization [22,23], Genetic algorithms [24–26], etc. A systematic review of global optimization methods can also be found in the work of Floudas et al. [27]. In addition, during the last few years, a variety of work has been proposed on combinations and modifications to some global optimization methods to more efficiently find the global minimum, such as methods that combine PSO with other methods [28–30], methods aimed to discover all the local minima of functions [31–33], new stopping rules to efficiently terminate the global optimization techniques [34–36], etc. In addition, due to the massive use of parallel processing techniques, several methods have been proposed that take full advantage of parallel processing, such as parallel techniques [37–39], methods that utilize the GPU architectures [40,41], etc.

In addition, during the past years, many metaheuristic algorithms have appeared to tackle global optimization problems such as the Quantum-based avian navigation optimizer

algorithm [42], a Tunicate Swarm Algorithm (TSA) inspired by simulating the lives of Tunicates at sea and how food is obtained [43], Starling murmuration optimizer [44,45], the Diversity-maintained multi-trial vector differential evolution algorithm (DMDE) used in large-scale global optimization [46], an improved moth-flame optimization algorithm with an adaptation mechanism to solve numerical and mechanical engineering problems [47], the dwarf-mongoose optimization algorithm [48], etc.

In this paper, a new multistart method is proposed that uses a machine learning model, which is trained in parallel with the evolution of the optimization process. Although multistart methods are considered the basis for more modern optimization techniques, they have been successfully used in several problems such as the Traveling Salesman Problem (TSP) [49–51], the maximum clique problem [52,53], the vehicle routing problem [54,55], scheduling problems [56,57], etc. In the new technique, a Radial Basis Function (RBF) network [58] is used to construct an approximation of the objective function. This construction is carried out in parallel with the execution of the optimization. A limited number of samples from the objective function and the local minima discovered during the optimization are used to construct the approximation function. During the execution of the method, the samples needed to start local minimizers are taken from the approximation function that is constructed by the neural network. The RBF network was used as an approximation tool as it has been successfully used in a wide range of problems in the field of artificial intelligence [59–62] and its training procedure is very fast, if compared to artificial neural networks, for example. In addition, for a more efficient termination of the method, a termination method proposed by Tsoulos is used [63], but this termination method is applied after each execution of the local minimization procedure. The mentioned method was applied to some test functions provided by the relevant literature and the results are extremely promising as compared with other global optimization techniques.

The proposed method does not sample the actual function but an approximation of it, which is generated incrementally. The creation of the approximation is done by using an RBF neural network, known for its reliability and its ability to efficiently approximate functions. The initial approximation is created from a limited number of points, and then, it will be improved through the local minimizers that will be found during the execution of the method. With the above procedure, the required number of function calls is drastically reduced, since the actual function is not used to produce samples, but an approximation of them. Only samples with low function values are taken from the approximation function, which means that finding the global minimum is likely to be performed faster than other techniques and more efficiently. Furthermore, the generation of the approximation function does not use any prior knowledge about the objective problem.

The rest of this article is organized as follows: in Section 2, the description of the proposed method is provided. In Section 3, the used experimental functions as well as the experimental results and comparisons are listed, and finally, in Section 4, some conclusions and final thoughts are given.

## 2. Method Description

The proposed technique generates an estimation of the objective function during the optimization using an RBF network. This estimation is initially generated from some samples from the objective function and gradually local minima that will have been discovered during the optimization are added to it. In this way, the estimation of the objective function will be continuously improved to approximate the true function as much as possible. At every iteration, several samples are then taken from the estimated function and sorted in ascending order. Those with the lowest value will be starting points of the local minimization method. The local optimization method used here is a BFGS variant of Powell [64]. This process has the effect of drastically reducing the total number of function calls that are made and, at the same time, the points used as initiators of the local minimization technique approach the global minimum of the objective function. In addition, the proposed method checks the termination rule after the application of every local search method. That way,

if the absolute minimum has already been discovered with some certainty, no more function calls will be wasted finding it.

In the following subsections, the training procedure of RBF networks as well as the proposed method are fully described.

*2.1. RBF Preliminaries*

An RBF network can be defined as:

$$N(\overrightarrow{x}) = \sum_{i=1}^{k} w_i \phi\left(\left\| \overrightarrow{x} - \overrightarrow{c_i} \right\|\right) \tag{2}$$

where

1.  The vector $\overrightarrow{x}$ is called the input pattern to the equation.
2.  The vectors $\overrightarrow{c_i}$, $i = 1, \ldots, k$ are called the center vectors.
3.  The vector $\overrightarrow{w}$ stands for the the output weight of the RBF network.

In most cases, the function $\phi(x)$ is a Gaussian function:

$$\phi(x) = \exp\left(-\frac{(x - c)^2}{\sigma^2}\right) \tag{3}$$

The training error for the RBF network on a set of points $T = \{(x_1, y_1), (x_2, y_2), \ldots, (x_M, y_M)\}$ is estimated as

$$E(N(\overrightarrow{x})) = \sum_{i=1}^{M} (N(x_i) - y_i)^2 \tag{4}$$

In most approaches, Equation (4) is minimized with respect to the parameters of the RBF network using a two-phase procedure:

1.  In the first phase, the K-Means algorithm [65] is used to approximate the $k$ centers and the corresponding variances.
2.  In the second phase, the weight vector $\overrightarrow{w} = (w_1, w_2, \ldots, w_k)$ is calculated by solving a linear system of equations as follows:
    (a)  **Set** $W = w_{kj}$.
    (b)  **Set** $\Phi = \phi_j(x_i)$.
    (c)  **Set** $T = \{t_i = f(x_i), i = 1, \ldots, M\}$.
    (d)  The system to be solved is identified as:

$$\Phi^T\left(T - \Phi W^T\right) = 0 \tag{5}$$

    With the solution:

$$W^T = \left(\Phi^T\Phi\right)^{-1}\Phi^T T = \Phi^\dagger T \tag{6}$$

*2.2. The Main Algorithm*

The main steps of the proposed algorithm are as follows:

1. **Initialization** step.
   - (a) **Set** $k$ the number of weights in the RBF network.
   - (b) **Set** $N_S$ the initial samples that will be taken from the function $f(x)$.
   - (c) **Set** $N_T$ the number of samples that will be used in every iteration as starting points for the local optimization procedure.
   - (d) **Set** $N_R$ the number of samples that will be drawn from the RBF network at every iteration with $N_R > N_T$.
   - (e) **Set** $N_G$ the maximum number of allowed iterations.
   - (f) **Set** Iter = 0, the iteration number.
   - (g) **Set** $(x^*, y^*)$ as the global minimum. Initially, $y^* = \infty$

2. **Creation** Step.
   - (a) **Set** $T = \emptyset$, the training set for the RBF network.
   - (b) **For** $i = 1, \ldots, N_S$ do
     - i. **Take** a new sample $x_i \in S$.
     - ii. **Calculate** $y_i = f(x_i)$.
     - iii. $T = T \cup (x_i, y_i)$.
   - (c) **End For**
   - (d) **Train** the RBF network on the training set $T$.

3. **Sampling** Step.
   - (a) **Set** $T_R = \emptyset$.
   - (b) **For** $i = 1, \ldots, N_R$ **do**
     - i. **Take** a random sample $(x_i, y_i)$ from the RBF network.
     - ii. **Set** $T_R = T_R \cup (x_i, y_i)$.
   - (c) **End For**
   - (d) **Sort** $T_R$ according to the y values in ascending order.

4. **Optimization** Step.
   - (a) **For** $i = 1, \ldots, N_T$ do
     - i. **Take** the next sample $(x_i, y_i)$ from $T_R$.
     - ii. $y_i = \text{LS}(x_i)$, where LS(x) is a predefined local search method.
     - iii. $T = T \cup (x_i, y_i)$; this step updates the training set of the RBF network.
     - iv. **Train** the RBF network on the set $T$.
     - v. **If** $y_i \leq y^*$, then $x^* = x_i, y^* = y_i$.
     - vi. **Check** the termination rule as suggested in [63]. If it holds, then report $(x^*, y^*)$ as the located global minimum and terminate.
   - (b) **End For**

5. **Set** iter = iter + 1.
6. **Goto** to Sampling step.

The steps of the algorithm are illustrated graphically in Figure 1.

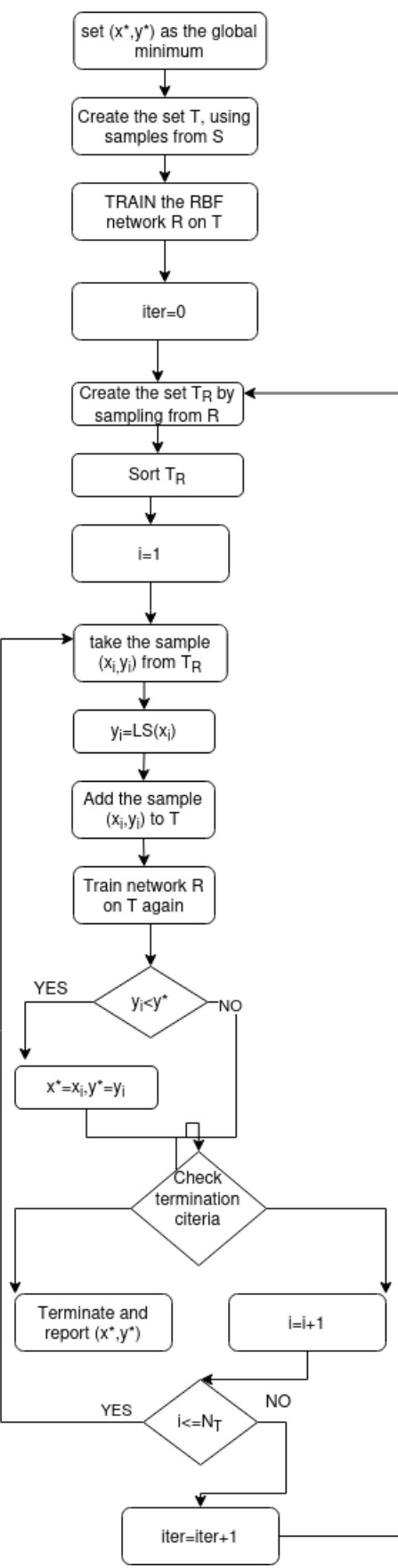

**Figure 1.** The steps of the proposed algorithm.

### 3. Experiments

To estimate the efficiency of the new technique, a number of functions from the relevant literature were used [66,67]. These functions are provided in Appendix A. The proposed technique was tested on these test functions and the results produced were compared with those given by a simple genetic algorithm or a PSO method or differential evolution (DE) method. The used genetic algorithm is based on the $GA(c_{r1}, l)$ algorithm from the work of Kaelo and Ali [68]. In order to have fairness in the comparison of the results, for all global optimization techniques, the same local minimization method as that of the proposed method has been used. In addition, the number of chromosomes in the genetic algorithm and the number of particles in the PSO method are identical to the parameter $N_T$ of the proposed procedure. In addition, for the DE method, the number of agents was set to $N_T$. The values for the parameters used in the conducted experiments are shown in Table 1. For every function and for every global optimizer, 30 independent runs were executed using a different seed for the random generator each time. The proposed method is implemented as the method with the name NeuralMinimizer in the OPTIMUS global optimization environment, which is freely available from https://github.com/itsoulos/OPTIMUS (accessed on 25 January 2023). All the experiments were conducted on an AMD Ryzen 5950X with 128 GB of RAM and the Debian Linux operating system.

**Table 1.** Experimental settings.

| Parameter | Meaning | Value |
|:---:|:---:|:---:|
| $k$ | Number of weights | 10 |
| $N_S$ | Start samples | 50 |
| $N_T$ | Number of samples used as starting points | 100 |
| $N_R$ | Number of samples that will be drawn from the RBF network | $10 \times N_T$ |
| $N_C$ | Chromosomes or Particles or agents | 100 |
| $N_G$ | Maximum number of iterations | 200 |

The experimental results from the application of the proposed method and the other methods are shown in Table 2. The number in the cells represent average function calls. The number in parentheses indicates the fraction of runs where the global optimum was successfully discovered. Absence of this fraction indicated that the global minimum is discovered for every execution (100% success). At the end of the table, an additional row named AVERAGE has been added to show the total number of function calls and the average success rate in locating the global minimum. In the experimental results, the superiority of the proposed technique over the other two methods in terms of the number of function calls is clear. The proposed technique requires an average of 90% fewer function calls than the other methods. In addition, the proposed technique appears to be more efficient than the other two as it finds, on average, more often the global minimum of most test functions in the experiments. In addition, the statistical comparison between the global optimization methods is shown in Figure 2.

**Table 2.** Comparison between the proposed method and the Genetic and PSO methods.

| Function | Genetic | PSO | DE | Proposed |
|---|---|---|---|---|
| BF1 | 7150 | 9030 (0.87) | 5579 | 1051 |
| BF2 | 7504 | 6505 (0.67) | 5598 | 921 |
| BRANIN | 6135 | 6865 (0.93) | 5888 | 460 |
| CAMEL | 6564 | 5162 | 6403 | 778 |
| CIGAR10 | 11,813 | 18,803 | 13,313 | 1896 |
| CM4 | 10,537 | 11,124 | 9018 | 1877 (0.87) |
| DISCUS10 | 20,208 | 6039 | 7797 | 478 |
| EASOM | 5281 | 2037 | 7917 | 258 |
| ELP10 | 20,337 | 16,731 | 2863 | 2263 |
| EXP4 | 10,537 | 9155 | 5944 | 750 |
| EXP16 | 20,131 | 14,061 | 3653 | 885 |
| EXP64 | 20,140 | 8958 | 3692 | 948 |
| GRIEWANK10 | 20,151 (0.10) | 17,497 (0.03) | 16,469 (0.03) | 2697 |
| POTENTIAL3 | 18,902 | 9936 | 5452 | 1192 |
| POTENTIAL5 | 18,477 | 12,385 | 3972 | 2399 |
| HANSEN | 10,708 | 9104 | 14,016 | 2370 (0.93) |
| HARTMAN3 | 8481 | 12,971 | 4677 | 642 |
| HARTMAN6 | 17,723 (0.60) | 15,174 (0.57) | 14,372 (0.90) | 883 |
| RASTRIGIN | 6744 | 7639 (0.97) | 6148 | 1408 (0.80) |
| ROSENBROCK4 | 20,815 (0.63) | 11,526 | 16,763 | 1619 |
| ROSENBROCK8 | 20,597 (0.67) | 16,967 | 16,631 | 2444 |
| SHEKEL5 | 14,456 (0.73) | 15,082 (0.47) | 13,178 | 2333 (0.87) |
| SHEKEL7 | 16,786 (0.83) | 14,625 (0.40) | 12,050 | 1844 (0.93) |
| SHEKEL10 | 15,586 (0.80) | 12,628 (0.53) | 13,107 | 2451 |
| SINU4 | 11,908 | 10,659 | 9048 | 802 |
| SINU8 | 20,115 | 13,912 | 16,210 | 1500 (0.97) |
| TEST2N4 | 13,943 | 12,948 | 10,864 | 878 (0.93) |
| TEST2N5 | 15,814 | 13,936 (0.90) | 15,259 | 971 (0.77) |
| TEST2N6 | 18,987 | 15,449 (0.70) | 12,839 | 997 (0.70) |
| TEST2N7 | 20,035 | 16,020 (0.50) | 8185 (0.97) | 1084 (0.30) |
| TEST30N3 | 13,029 | 7239 | 4839 | 1061 |
| TEST30N4 | 12,889 | 8051 | 5070 | 854 |
| **Average** | **472,596 (0.89)** | **368,218 (0.86)** | **296,814 (0.96)** | **42,994 (0.94)** |

In addition, Table 3 presents the experimental results for the proposed method and for various values of the parameter $N_S$. As can be seen, the increase in this parameter does not cause a large increase in the total number of function calls, while, at the same time, it improves to some extent the ability of the proposed technique to find the global minimum.

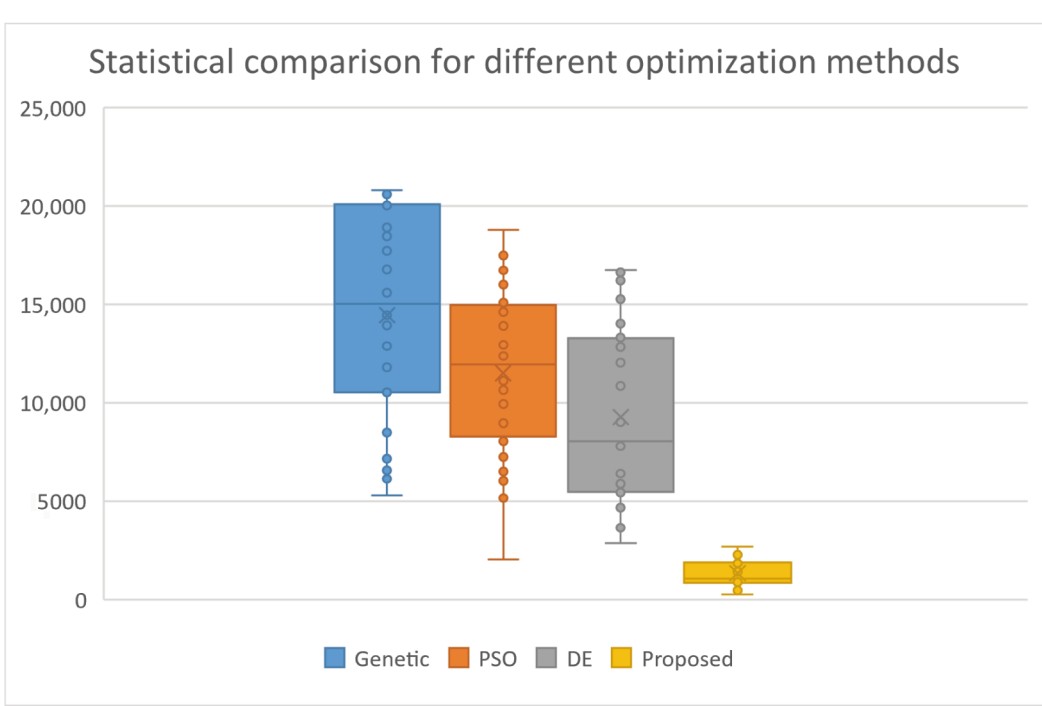

**Figure 2.** Statistical comparison between the global optimization methods.

**Table 3.** Experimental results for the proposed method and for different values of the critical parameter $N_S$ (50, 100, 200). Numbers in the cells represent averages of 30 runs.

| Function | $N_S = 50$ | $N_S = 100$ | $N_S = 200$ |
|---|---|---|---|
| BF1 | 1051 | 1116 | 1224 |
| BF2 | 921 | 949 | 1058 |
| BRANIN | 460 | 506 | 599 |
| CAMEL | 778 | 676 | 739 |
| CIGAR10 | 1896 | 1934 | 2042 |
| CM4 | 1877 (0.87) | 1859 (0.93) | 1877 (0.90) |
| DISCUS10 | 478 | 531 | 634 |
| EASOM | 258 | 307 | 450 |
| ELP10 | 2263 | 2339 | 3130 |
| EXP4 | 750 | 778 | 884 |
| EXP16 | 885 | 932 | 1030 |
| EXP64 | 948 | 998 | 1091 |
| GRIEWANK10 | 2697 | 2647 | 2801 |
| POTENTIAL3 | 1192 | 1228 | 1305 |
| POTENTIAL5 | 2399 | 2417 | 2544 |
| HANSEN | 2370 (0.93) | 2602 (0.93) | 2578 (0.97) |
| HARTMAN3 | 642 | 696 | 798 |
| HARTMAN6 | 883 | 940 | 1038 |
| RASTRIGIN | 1408 (0.80) | 989 (0.83) | 1041 |
| ROSENBROCK4 | 1619 | 1674 | 1751 |
| ROSENBROCK8 | 2444 | 2499 | 2583 |
| SHEKEL5 | 2333 (0.87) | 1267 | 1878 (0.97) |

**Table 3.** *Cont.*

| Function | $N_S = 50$ | $N_S = 100$ | $N_S = 200$ |
|:---:|:---:|:---:|:---:|
| SHEKEL7 | 1844 (0.93) | 1517 (0.93) | 1685 (0.97) |
| SHEKEL10 | 2451 | 2695 | 1498 |
| SINU4 | 802 | 821 | 901 |
| SINU8 | 1500 (0.97) | 1216 | 1247 |
| TEST2N4 | 878 (0.93) | 934 | 850 (0.97) |
| TEST2N5 | 971 (0.77) | 941 (0.80) | 993 |
| TEST2N6 | 997 (0.70) | 1087 (0.77) | 1098 |
| TEST2N7 | 1084 (0.30) | 1160 (0.53) | 1313 (0.57) |
| TEST30N3 | 1061 | 998 | 1320 |
| TEST30N4 | 854 | 830 | 1108 |
| **Average** | **42,994 (0.94)** | **42,083 (0.96)** | **45,088 (0.97)** |

The efficiency of the method is also shown in Table 4, where the proposed method is compared against the genetic algorithm and particle swarm optimization for a range of number of atoms of the Potential problem. As can be seen in the table, the proposed method requires a significantly smaller number of function calls compared with the other techniques and its reliability in finding the global minimum remains high even when the number of atoms in the potential increases significantly.

**Table 4.** Optimizing the Potential problem for different number of atoms.

| Atoms | Genetic | PSO | Proposed |
|:---:|:---:|:---:|:---:|
| 3 | 18,902 | 9936 | 1192 |
| 4 | 17,806 | 12,560 | 1964 |
| 5 | 18,477 | 12,385 | 2399 |
| 6 | 19,069 (0.20) | 9683 | 3198 |
| 7 | 16,390 (0.33) | 10,533 (0.17) | 3311 (0.97) |
| 8 | 15,924 (0.50) | 8053 (0.50) | 3526 |
| 9 | 15,041 (0.27) | 9276 (0.17) | 4338 |
| 10 | 14,817 (0.03) | 7548 (0.17) | 5517 (0.87) |
| 11 | 13,885 (0.03) | 6864 (0.13) | 6588 (0.80) |
| 12 | 14,435 (0.17) | 12,182 (0.07) | 7508 (0.83) |
| 13 | 14,457 (0.07) | 10,748 (0.03) | 6717 (0.77) |
| 14 | 13,906 (0.07) | 14,235 (0.13) | 6201 (0.93) |
| 15 | 12,832 (0.10) | 12,980 (0.10) | 7802 (0.90) |
| **Average** | **205,941 (0.37)** | **137,134 (0.42)** | **60,258 (0.93)** |

## 4. Conclusions

An innovative technique for finding the global minimum of multidimensional functions was presented in this work. This new technique is based on the multistart procedure, but also generates an estimation of the objective function through a machine learning model. The machine learning model constructs an estimation of the objective function using a small number of samples from the true function but also with the contribution of local minima discovered during the execution of the method. In this way, the estimation of the objective function is continuously improved and the sampling to perform local minimization is done

from the estimated function rather than the actual one. This procedure combined with checking the termination criterion after each execution of the local minimization method led the proposed method to have excellent results both in terms of the speed of finding the global minimum and its efficiency. In addition, the method shows significant stability in its performance even in the presence of large changes of its parameters as presented in the experimental results section.

In the future, the use of the RBF network to construct an approximation of the objective function can be applied to more modern optimization techniques such as genetic algorithms. It would also be interesting to create a parallel implementation of the proposed method, in order to significantly speed up its execution and to be able to be used efficiently in optimization problems of higher dimensions.

**Author Contributions:** I.G.T., A.T., E.K. and D.T. conceived of the idea and methodology. I.G.T. and A.T. conducted the experiments, employing several test functions and provided the comparative experiments. E.K. and D.T. performed the statistical analysis and all other authors prepared the manuscript. All authors have read and agreed to the published version of the manuscript.

**Funding:** This research received no external funding.

**Conflicts of Interest:** The authors declare no conflict of interest.

## Appendix A

- **Bent Cigar function.** The function is

$$f(x) = x_1^2 + 10^6 \sum_{i=2}^{n} x_i^2$$

  with the global minimum $f(x^*) = 0$. For the conducted experiments, the value $n = 10$ was used.

- **Bf1** function. The function Bohachevsky 1 is given by the equation

$$f(x) = x_1^2 + 2x_2^2 - \frac{3}{10}\cos(3\pi x_1) - \frac{4}{10}\cos(4\pi x_2) + \frac{7}{10}$$

  with $x \in [-100, 100]^2$.

- **Bf2** function. The function Bohachevsky 2 is given by the equation

$$f(x) = x_1^2 + 2x_2^2 - \frac{3}{10}\cos(3\pi x_1)\cos(4\pi x_2) + \frac{3}{10}$$

  with $x \in [-50, 50]^2$.

- **Branin** function. The function is defined by $f(x) = \left(x_2 - \frac{5.1}{4\pi^2}x_1^2 + \frac{5}{\pi}x_1 - 6\right)^2 + 10\left(1 - \frac{1}{8\pi}\right)\cos(x_1) + 10$ with $-5 \le x_1 \le 10$, $0 \le x_2 \le 15$. The value of global minimum is 0.397887 with $x \in [-10, 10]^2$.

- **CM** function. The Cosine Mixture function is given by the equation

$$f(x) = \sum_{i=1}^{n} x_i^2 - \frac{1}{10}\sum_{i=1}^{n}\cos(5\pi x_i)$$

  with $x \in [-1, 1]^n$. For the conducted experiments, the value $n = 4$ was used.

- **Camel** function. The function is given by

$$f(x) = 4x_1^2 - 2.1x_1^4 + \frac{1}{3}x_1^6 + x_1 x_2 - 4x_2^2 + 4x_2^4, \quad x \in [-5, 5]^2$$

  The global minimum has the value of $f(x^*) = -1.0316$.

- **Discus function.** The function is defined as

$$f(x) = 10^6 x_1^2 + \sum_{i=2}^{n} x_i^2$$

with global minimum $f(x^*) = 0$. For the conducted experiments, the value $n = 10$ was used.

- **Easom** function. The function is given by the equation

$$f(x) = -\cos(x_1)\cos(x_2)\exp\left((x_2 - \pi)^2 - (x_1 - \pi)^2\right)$$

with $x \in [-100, 100]^2$ and global minimum $-1.0$.

- **Exponential** function.
The function is given by

$$f(x) = -\exp\left(-0.5\sum_{i=1}^{n} x_i^2\right), \quad -1 \le x_i \le 1$$

The values $n = 4, 16, 64$ were used here and the corresponding function names are EXP4, EXP16, EXP64.

- **Griewank10** function, defined as:

$$f(x) = \sum_{i=1}^{n} \frac{x_i^2}{4000} - \prod_{i=1}^{n} \cos\left(\frac{x_i}{\sqrt{i}}\right) + 1$$

with $n = 10$.

- **Hansen** function. $f(x) = \sum_{i=1}^{5} i\cos[(i-1)x_1 + i]\sum_{j=1}^{5} j\cos[(j+1)x_2 + j]$, $x \in [-10, 10]^2$. The global minimum of the function is $-176.541793$.

- **Hartman 3** function. The function is given by

$$f(x) = -\sum_{i=1}^{4} c_i \exp\left(-\sum_{j=1}^{3} a_{ij}(x_j - p_{ij})^2\right)$$

with $x \in [0, 1]^3$ and $a = \begin{pmatrix} 3 & 10 & 30 \\ 0.1 & 10 & 35 \\ 3 & 10 & 30 \\ 0.1 & 10 & 35 \end{pmatrix}$, $c = \begin{pmatrix} 1 \\ 1.2 \\ 3 \\ 3.2 \end{pmatrix}$ and

$$p = \begin{pmatrix} 0.3689 & 0.117 & 0.2673 \\ 0.4699 & 0.4387 & 0.747 \\ 0.1091 & 0.8732 & 0.5547 \\ 0.03815 & 0.5743 & 0.8828 \end{pmatrix}$$

The value of the global minimum is $-3.862782$.

- **Hartman 6** function.

$$f(x) = -\sum_{i=1}^{4} c_i \exp\left(-\sum_{j=1}^{6} a_{ij}(x_j - p_{ij})^2\right)$$

with $x \in [0, 1]^6$ and $a = \begin{pmatrix} 10 & 3 & 17 & 3.5 & 1.7 & 8 \\ 0.05 & 10 & 17 & 0.1 & 8 & 14 \\ 3 & 3.5 & 1.7 & 10 & 17 & 8 \\ 17 & 8 & 0.05 & 10 & 0.1 & 14 \end{pmatrix}$, $c = \begin{pmatrix} 1 \\ 1.2 \\ 3 \\ 3.2 \end{pmatrix}$ and

$$p = \begin{pmatrix} 0.1312 & 0.1696 & 0.5569 & 0.0124 & 0.8283 & 0.5886 \\ 0.2329 & 0.4135 & 0.8307 & 0.3736 & 0.1004 & 0.9991 \\ 0.2348 & 0.1451 & 0.3522 & 0.2883 & 0.3047 & 0.6650 \\ 0.4047 & 0.8828 & 0.8732 & 0.5743 & 0.1091 & 0.0381 \end{pmatrix}$$

The value of the global minimum is $-3.322368$.

- **High Conditioned Elliptic** function, defined as

$$f(x) = \sum_{i=1}^{n} \left(10^6\right)^{\frac{i-1}{n-1}} x_i^2$$

with $n = 10$ for the conducted experiments.

- **Potential** function used to represent the lowest energy for the molecular conformation of N atoms via the Lennard–Jones potential [69]. The function is defined as:

$$V_{LJ}(r) = 4\epsilon \left[ \left(\frac{\sigma}{r}\right)^{12} - \left(\frac{\sigma}{r}\right)^6 \right] \tag{A1}$$

In the current experiments, two different cases were studied: $N = 3, 5$.

- **Rastrigin** function. The function is given by

$$f(x) = x_1^2 + x_2^2 - \cos(18x_1) - \cos(18x_2), \quad x \in [-1, 1]^2$$

- **Shekel 7** function.

$$f(x) = -\sum_{i=1}^{7} \frac{1}{(x - a_i)(x - a_i)^T + c_i}$$

with $x \in [0, 10]^4$ and $a = \begin{pmatrix} 4 & 4 & 4 & 4 \\ 1 & 1 & 1 & 1 \\ 8 & 8 & 8 & 8 \\ 6 & 6 & 6 & 6 \\ 3 & 7 & 3 & 7 \\ 2 & 9 & 2 & 9 \\ 5 & 3 & 5 & 3 \end{pmatrix}$, $c = \begin{pmatrix} 0.1 \\ 0.2 \\ 0.2 \\ 0.4 \\ 0.4 \\ 0.6 \\ 0.3 \end{pmatrix}$.

- **Shekel 5** function.

$$f(x) = -\sum_{i=1}^{5} \frac{1}{(x - a_i)(x - a_i)^T + c_i}$$

with $x \in [0, 10]^4$ and $a = \begin{pmatrix} 4 & 4 & 4 & 4 \\ 1 & 1 & 1 & 1 \\ 8 & 8 & 8 & 8 \\ 6 & 6 & 6 & 6 \\ 3 & 7 & 3 & 7 \end{pmatrix}$, $c = \begin{pmatrix} 0.1 \\ 0.2 \\ 0.2 \\ 0.4 \\ 0.4 \end{pmatrix}$.

- **Shekel 10** function.

$$f(x) = -\sum_{i=1}^{10} \frac{1}{(x - a_i)(x - a_i)^T + c_i}$$

with $x \in [0, 10]^4$ and $a = \begin{pmatrix} 4 & 4 & 4 & 4 \\ 1 & 1 & 1 & 1 \\ 8 & 8 & 8 & 8 \\ 6 & 6 & 6 & 6 \\ 3 & 7 & 3 & 7 \\ 2 & 9 & 2 & 9 \\ 5 & 5 & 3 & 3 \\ 8 & 1 & 8 & 1 \\ 6 & 2 & 6 & 2 \\ 7 & 3.6 & 7 & 3.6 \end{pmatrix}$, $c = \begin{pmatrix} 0.1 \\ 0.2 \\ 0.2 \\ 0.4 \\ 0.4 \\ 0.6 \\ 0.3 \\ 0.7 \\ 0.5 \\ 0.6 \end{pmatrix}$.

- **Sinusoidal** function. The function is given by

$$f(x) = -\left( 2.5 \prod_{i=1}^{n} \sin(x_i - z) + \prod_{i=1}^{n} \sin(5(x_i - z)) \right), \quad 0 \le x_i \le \pi.$$

The global minimum is located at $x^* = (2.09435, 2.09435, \ldots, 2.09435)$ with $f(x^*) = -3.5$. For the conducted experiments, the cases of $n = 4, 8$, and $z = \frac{\pi}{6}$ were studied.

- **Test2N** function. This function is given by the equation

$$f(x) = \frac{1}{2} \sum_{i=1}^{n} x_i^4 - 16x_i^2 + 5x_i, \quad x_i \in [-5, 5].$$

The function has $2^n$ local minima in the specified range and, in our experiments, we used $n = 4, 5, 6, 7$.

- **Test30N** function. This function is given by

$$f(x) = \frac{1}{10} \sin^2(3\pi x_1) \sum_{i=2}^{n-1} \left( (x_i - 1)^2 \left( 1 + \sin^2(3\pi x_{i+1}) \right) \right) + (x_n - 1)^2 \left( 1 + \sin^2(2\pi x_n) \right)$$

with $x \in [-10, 10]$. The function has $30^n$ local minima in the specified range and we used $n = 3, 4$ in the conducted experiments.

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
