# Peer review of "NeuralMinimizer: A Novel Method for Global Optimization"

_information, doi:10.3390/info14020066_

Round 1

Reviewer 1 Report

The authors present a topic of interest, the search for the global minimum of the multidimensional function with applicability in Machine Learning, using an RBF type neural network. From a mathematical point of view, the proposed algorithm is proven and stable, but which Machine Learning model is proposed?

I recommend the authors to exemplify the proposed Machine Learning model (diagram) because the RBF neural network has been used in the specialized literature for such applications.

Possibly the authors should propose in the article an applicability for the proposed algorithm, possibly jointly with other topics on this topic in the literature, so that the performance of the proposed algorithm on a certain topic results, not only mathematically at the demonstration level.

Author Response

1. COMMENT

The authors present a topic of interest, the search for the global minimum of the multidimensional function with applicability in Machine Learning, using an RBF type neural network. From a mathematical point of view, the proposed algorithm is proven and stable, but which Machine Learning model is proposed?

RESPONSE

The main title has been simplified to “NeuralMinimizer, a novel method for global optimization”

2. COMMENT

I recommend the authors to exemplify the proposed Machine Learning model (diagram) because the RBF neural network has been used in the specialized literature for such applications.

RESPONSE

A diagram of the proposed method has been added at the end of subsection 2.2

3. COMMENT

Possibly the authors should propose in the article an applicability for the proposed algorithm, possibly jointly with other topics on this topic in the literature, so that the performance of the proposed algorithm on a certain topic results, not only mathematically at the demonstration level.

RESPONSE

1)The following paragraph has been added at the Introduction section:

Also, during the past years many metaheuristic algorithms appeared to tackle to global optimization problems such as Quantum-based avian navigation optimizer algorithm [qana], a Tunicate Swarm Algorithm (TSA) inspired by simulating the lives of Tunicates at sea and how food is obtained [tsa], Starling murmuration optimizer [starling1, starling2], the Diversity-maintained multi-trial vector differential evolution algorithm (DMDE) algorithm used in large scale global optimization [dmde], an improved moth-flame optimization algorithm with adaptation mechanism to solve numerical and mechanical engineering problems [flame], the dwarf mongoose optimization algorithm [dwarf] etc.

2)The proposed method was compared against Genetic Algorithm and PSO for different number of atoms for the Potential problem in order to show its efficiency and a new table with experimental results has been added. Also the added text reads:

The efficiency of the method is also shown in Table tab:potential, where the proposed method is compared against genetic algorithm and particle swarm optimization for a range of number of atoms of the Potential problem. As can be seen in the table, the proposed method requires a significantly smaller number of function calls compared to the other techniques and its reliability in finding the global minimum remains high even when the number of people in the potential increases significantly.”

Reviewer 2 Report

1. The novelty of this study is not significant enough. It is recommended to revise the proposed method section.

2. It is recommended to review the abstract and highlight the novelty and findings of this study. A summary of the proposed method is needed.

3. Contributions of this study should be listed at the end of the introduction.

The importance of this study should be stated in the introduction, which can answer the following question: Why is this new algorithm needed, and what are the strengths of this new algorithm than the existing algorithms?

4. This study suffers from a lack of literature review of the problem of this study. It is recommended to add a related work section. Accordingly, the literature should describe the advantage of the proposed algorithm over other metaheuristic algorithms such as QANA: Quantum-based avian navigation optimizer algorithm, Starling murmuration optimizer, DMDE: Diversity-maintained multi-trial vector differential evolution algorithm for non-decomposition large-scale global optimization, and An improved moth-flame optimization algorithm with adaptation mechanism to solve numerical and mechanical engineering problems.

5. It is recommended to check the upper and lower cases in the whole manuscript. For example, 2.1 Rbf preliminaries.

6. Figure 1: A plot for the Gaussian function is needed?

7. It is recommended to consider the detail of the test function in the appendix.

8. The visualization of this study should be boosted. The convergence curve, box plot and etc are recommended.

9. The proposed algorithm was compared with PSO and Genetic. It is not clear how the Genetic algorithm as a discrete optimizer behaves to solve a continuous problem, and more detail is needed.

10. Comparing the proposed algorithm with the well-known and recent optimizers is suggested.

11. Experimental evaluations have not supported the claims of this study. Please clarify and boost this section.

Author Response

1. COMMENT

It is recommended to review the abstract and highlight the novelty and findings of this study. A summary of the proposed method is needed.

RESPONSE

The abstract has been enchanced to the following:

The problem of finding the global minimum of multidimensional functions is often applied to a wide range of problems. An innovative method of finding the global minimum of multidimensional functions is presented here. This method first generates an approximation of the objective function using only a few real samples from it. These samples construct the approach using a machine learning model. Next, the required sampling is performed by the approximation function. Furthermore, the approach is improved on each sample by using found local minima as samples for the training set of the machine learning model. In addition, the proposed technique uses as a termination criterion a widely used criterion from the relevant literature which in fact evaluates it after each execution of the local minimization. The proposed technique was applied to a number of well-known problems from the relevant literature, and the comparative results with respect to modern global minimization techniques are shown extremely promising.”

2. COMMENT

Contributions of this study should be listed at the end of the introduction. The importance of this study should be stated in the introduction, which can answer the following question: Why is this new algorithm needed, and what are the strengths of this new algorithm than the existing algorithms?

RESPONSE

The following paragraph has been added at the end of the Introduction section:

The proposed method does not sample the actual function but an approximation of it, which is generated incrementally. The creation of the approximation is done by using an RBF neural network, known for its reliability and its ability to efficiently approximate functions. The initial approximation is created from a limited number of points and then it will be improved through the local minimizers that will be found during the execution of the method. With the above procedure, the required number of function calls is drastically reduced, since the actual function is not used to produce samples, but an approximation of it. Only samples with low function values are taken from the approximation function, which means that finding the global minimum is likely to be performed faster than other techniques and more efficiently. Furthermore, the generation of the approximation function does not use any prior knowledge about the objective problem.

3. COMMENT

This study suffers from a lack of literature review of the problem of this study. It is recommended to add a related work section. Accordingly, the literature should describe the advantage of the proposed algorithm over other metaheuristic algorithms such as QANA: Quantum-based avian navigation optimizer algorithm, Starling murmuration optimizer, DMDE: Diversity-maintained multi-trial vector differential evolution algorithm for non-decomposition large-scale global optimization, and An improved moth-flame optimization algorithm with adaptation mechanism to solve numerical and mechanical engineering problems.

RESPONSE

The following paragraph has been added at the Introduction section:

Also, during the past years many metaheuristic algorithms appeared to tackle to global optimization problems such as Quantum-based avian navigation optimizer algorithm [qana], a Tunicate Swarm Algorithm (TSA) inspired by simulating the lives of Tunicates at sea and how food is obtained [tsa], Starling murmuration optimizer [starling1, starling2], the Diversity-maintained multi-trial vector differential evolution algorithm (DMDE) algorithm used in large scale global optimization [dmde], an improved moth-flame optimization algorithm with adaptation mechanism to solve numerical and mechanical engineering problems [flame], the dwarf mongoose optimization algorithm [dwarf] etc.

4. COMMENT

It is recommended to check the upper and lower cases in the whole manuscript. For example, 2.1 Rbf preliminaries.

RESPONSE

Done.

5. COMMENT

Figure 1: A plot for the Gaussian function is needed?

RESPONSE

Removed.

6. COMMENT

It is recommended to consider the detail of the test function in the appendix.

RESPONSE

The test functions have been moved to the Appendix A.

7. COMMENT

The visualization of this study should be boosted. The convergence curve, box plot and etc are recommended.

RESPONSE

Comparison between the different global optimization methods is added using boxplots.

8. COMMENT

The proposed algorithm was compared with PSO and Genetic. It is not clear how the Genetic algorithm as a discrete optimizer behaves to solve a continuous problem, and more detail is needed.

RESPONSE

We have added the following paragraph in Experiments section to clarify this:

The used genetic algorithm is based on the GA ( c r1 ,l ) algorithm from the work of Kaelo and Ali [kaelo].

9. COMMENT

Comparing the proposed algorithm with the well-known and recent optimizers is suggested.

RESPONSE

The method was also compared against the Differential Evolution (DE) method and the results have been added to the corresponding table.

Reviewer 3 Report

I suggest that title should be changed:
Neural Minimizer - a novel method for global optimization 
(I think that last 4 words are not necesary).

Abstract is OK, sufficient. I would change last two words in abstract, instead 
"optimization techniques are extremely promising"
put
"optimization techniques are shown".

Paper consists of sections:
1. Introduction
2.  Method description
    2.1. Rbf preliminaries
    2.2. The main algorithm
3. Experiments
    3.1. Test functions
    3.2. Experimental results 
4. Conclusions
References

Introduction is good and sufficient. Maybe to long, but sufficient. Referencing is good.

IN section 2
row 90, Rbf should br RBF
row 92, instead "Where" to put "where"
row 100, "equation 4", is maybe better to be "equation (4)"
why numerating formulaes 1,2,3,5,6 if authors do not reference to any of them?
rows 113-150, algorithm is written in not so good manner... it looks a little bit ugly. I suggest authors to try to make it "easier for reading".

I suggest (not obligatory) that authors that reorganize this section and subsections
2.  Method description
    2.1. Rbf preliminaries
    2.2. The main algorithm

to be only

2. Method and algorithm 

In section 3:
I do not understand why section 3.1 and listing of so many functions. Is it possible that those function are introduced as references in some other papers? Or if not, then to explain hem a little bit better (or to put citation with each of them)?
I do not understand notation [-100,100]2 
What means red number?

Subsection with results is OK. Table 2. clear and concise.
I do not understand Table 3, what is in it? Which algorithm is used?

I suggest (not obligatory) reorganization of section and subsections, instead:

3. Experiments
    3.1. Test functions
    3.2. Experimental results

to be just

3. Experimental results

Section Conclusion is OK, nice and clear.
References also good.

At the end...
I have to admit that I still do not understand purpose of this paper. I do not see clear what are new results. If it is an algorithm (i suppose), then it should be shown a little bit more explicit.

Maybe to add some analyses after tables 2 and 3, to explain why results are better, what is main contribution of new algorithm.

With those minor changes, this paper could be accepted and published,

Author Response

1. COMMENT

I suggest that title should be changed: Neural Minimizer - a novel method for global optimization (I think that last 4 words are not necesary).

RESPONSE

Done.

2. COMMENT

Abstract is OK, sufficient. I would change last two words in abstract, instead "optimization techniques are extremely promising put "optimization techniques are shown".

RESPONSE

Done.

3. COMMENT

IN section 2

  • row 90, Rbf should br RBF

  • row 92, instead "Where" to put "where"

RESPONSE

Done.

4. COMMENT

In section 3: I do not understand why section 3.1 and listing of so many functions. Is it possible that those function are introduced as references in some other papers? Or if not, then to explain hem a little bit better (or to put citation with each of them)?

RESPONSE

The test functions have been moved to the Appendix A.

5. COMMENT

I do not understand notation [-100,100]2  What means red number?

RESPONSE

Maybe a typo from the MDPI latex.

6. COMMENT

I do not understand Table 3, what is in it? Which algorithm is used?

RESPONSE

The caption in table have been changed to the following:

Experimental results for the proposed method and for different values of the critical parameter N_S (50, 100, 200). Numbers in cells represent averages of 30 runs.

7. COMMENT

I suggest (not obligatory) reorganization of section and subsections, instead:

3. Experiments
    3.1. Test functions
    3.2. Experimental results

to be just

3. Experimental results

RESPONSE

Done.

8. COMMENT

I have to admit that I still do not understand purpose of this paper. I do not see clear what are new results. If it is an algorithm (i suppose), then it should be shown a little bit more explicit. Maybe to add some analyses after tables 2 and 3, to explain why results are better, what is main contribution of new algorithm.

RESPONSE

1)The following paragraph has been added at the end of the Introduction section:

The proposed method does not sample the actual function but an approximation of it, which is generated incrementally. The creation of the approximation is done by using an RBF neural network, known for its reliability and its ability to efficiently approximate functions. The initial approximation is created from a limited number of points and then it will be improved through the local minimizers that will be found during the execution of the method. With the above procedure, the required number of function calls is drastically reduced, since the actual function is not used to produce samples, but an approximation of it. Only samples with low function values are taken from the approximation function, which means that finding the global minimum is likely to be performed faster than other techniques and more efficiently. Furthermore, the generation of the approximation function does not use any prior knowledge about the objective problem.

2)Comparison between the different global optimization methods is added using boxplots.

Round 2

Reviewer 1 Report

The authors responded to the clarifications.

Author Response

1. Comment

The authors responded to the clarifications.

Response

Dear reviewer thank you for your valuable time.

Reviewer 2 Report

The authors have responded to my comments adequately. However, the author should check all cross-references in the content carefully before publishing because they have been wrongly replaced with the question mark. 

Author Response

1. Comment

The authors have responded to my comments adequately. However, the author should check all cross-references in the content carefully before publishing because they have been wrongly replaced with the question mark.

Response

Thank you for your comment. It was a problem with Latex typesetting system.